# "Let's see what happens:"—Women's experiences of open-label placebo treatment for menopausal hot flushes in a randomized controlled trial

Yiqi Pan[1]*, Miriam L. Frank[1], Ted J. Kaptchuk[2], Yvonne Nestoriuc[3,4]

1 Department of Psychosomatic Medicine and Psychotherapy, University Medical Centre Hamburg-Eppendorf, Hamburg, Hamburg, Germany, 2 Program in Placebo Studies and the Therapeutic Encounter (PiPS), Beth Israel Deaconess Medical Center, Harvard Medical School, Boston, Massachusetts, United States of America, 3 Clinical Psychology, Helmut Schmidt University/University of the Federal Armed Forces Hamburg, Hamburg, Hamburg, Germany, 4 Institute for Systems Neuroscience, University Medical Centre Hamburg-Eppendorf, Hamburg, Hamburg, Germany

* y.pan@uke.de

## Abstract

Open-label (honestly prescribed) placebos are an ethical way to evoke placebo effects in patients. As part of a mixed-methods study, we conducted in-depth interviews with eight menopausal women who underwent and benefitted from open-label placebo treatment in a randomized-controlled trial of hot flushes. Data were analyzed using Interpretative Phenomenological Analysis. We found that the women had low expectations about the placebo treatment yet endorsed what they referred to as "hope" and openness to "see what happens". Recording hot flushes via the symptom diary was viewed as a valuable opportunity for self-examination and appraising outcomes. Receiving relief from the placebo treatment empowered women and enhanced their sense of control and agency. In summary, participants' initial openness towards placebos, their hopes to get better, monitoring symptoms closely, and taking the initiative to address symptoms were components of a positive open-label placebo experience.

## Introduction

The relationship between biomedicine and placebo effects has been a convoluted one [1]. On the one hand side, placebo effects are viewed as a nuisance that hinders isolation of the specific components of the treatment, for instance, the pharmacological effects of a medication in randomized controlled trials (RCTs) [2]. On the other hand, placebo treatment seems to offer salubrious effects [3]. Recent research into the neurobiological underpinnings of placebo effects has conferred increased respectability to the phenomenon [4–6]. For example, specific neurotransmitters (e.g., endorphins, dopamine and, cannabinoids) are involved, and quantifiable and relevant areas of the brain are engaged in placebo effects (e.g., prefrontal cortex, anterior insula, rostral anterior cingulate cortex) [7]. Potential genetic signatures of placebo effects

**Data Availability Statement:** All relevant data are within the paper and its Supporting Information

files. However, full interviews were not included to protect the confidentiality of the participants.

**Funding:** The study was supported by the Foundation for the Science of the Therapeutic Encounter (F-STE) under grant number 1705-100 (recipient: YN). The funder had no role in study design, data collection, analysis, decision to publish, or preparation of the manuscript. Website of the funder: https://www.stefound.org/. It was also supported by the Deutsche Forschungsgemeinschaft (DFG, German Research Foundation): TRR 289 Treatment Expectation— Project Number 422744262 (Recipient: YN).

**Competing interests:** The authors have declared that no competing interests exist.

have also been detected [8]. Furthermore, placebo treatments demonstrated the ability to improve subjective symptoms in multiple conditions [3, 9]. Such findings spurred the scientific communities' renewed interest in placebo effects as powerful manifestations of the mind-body connection.

As knowledge about the clinical implications of placebo effects and their biological under-pinnings accumulated, efforts to harness placebo treatments have been hindered by the belief that placebos require deception and concealment to trigger benefits. For clinical care, such deception or concealment is ethically problematic. In 2010, this belief of required "patient ignorance" was challenged by a landmark RCT demonstrating that improvement could be evoked even if patients were honestly informed about the pills' lack of active substance [10]. Since then, at least thirteen RCTs comparing OLP to no-treatment have been performed in various patient groups [11], including irritable bowel syndrome (IBS), episodic migraine attacks, chronic low back pain, and cancer-related fatigue. A meta-analysis demonstrated that OLP treatment could produce statistically significant and clinically meaningful improvements across various illnesses defined by subjective complaints [11].

In this study, we investigated the experiences of menopausal women who underwent and benefitted from OLP treatment for hot flushes that was administered as part of an RCT. Our primary interest was exploring each participant's individual treatment experiences in light of their symptom history and attitudes towards their symptoms. Generating quantitative and qualitative data about the OLP treatment might have clinical relevance for menopausal women, given the current lack of effective hot flush treatments with high patient acceptance. Many women are concerned about the potential long-term adverse effects of the gold standard hormone therapy [12], and the demand for complementary and alternative medicine (CAM) treatments is high [13]. However, the evidence base for such treatments, inter alia, black cohosh, soy isoflavones, Chinese herbs, has been mixed [14]. Relevantly, the improvements in the placebo arm were shown to be large in double-blind RCTs [15]. In the preceding RCT, we found that honestly prescribed placebos significantly reduced hot flush symptoms and improved menopause-related quality of life compared to no-treatment, with moderate to large effect sizes [16]. In this qualitative study, we discern these developments from the patients' perspective.

## Materials and methods

### Design

This study is the second part of an explanatory sequential mixed-methods design. Details on the design of the preceding quantitative study, i.e., the RCT, are provided in the study protocol [17] and the RCT results publication [16]. In brief, 100 women in peri- or post-menopause who had at least five hot flushes per day and were moderately or severely burdened by their symptoms were randomized to receive four weeks of OLP or no treatment. Before the randomization, all participants protocoled their hot flushes for a week (baseline assessment). After completing the treatment for four weeks, the OLP group was allocated a second time to discontinue or continue the treatment for another four weeks. The treatment consisted of a twice-daily intake of white, uncoated placebo pills for four or eight weeks and four study visits consisting of questionnaire assessments and short conversations with a clinician. At the first study visit, all participants were informed about why the placebo might be effective (see S1 Appendix for the placebo rationale). All women completed a paper-pencil diary in which they indicated the frequency and severity of each hot flush. The study was approved by the ethics committee of the medical chamber in Hamburg (trial number: PV5787) and pre-registered at

ClinicalTrials.gov (ID: NCT03838523). We adhere to the Consolidated Criteria for Reporting Qualitative Research (COREQ) guideline when reporting the study [18].

As specified in the study protocol [17], interview conceptualization, data analysis, and interpretation were conducted using Interpretative Phenomenological Analysis (IPA). We chose this approach as phenomenology is well-suited to describe the individual's meaning of experiences [18]. IPA accentuates that the researcher should strive to understand the participant's experiences in the context of the participant's realities. Researchers ought to engage in the double hermeneutic process, "whereby the researcher is trying to make sense of the participant trying to make sense of what is happening to them." [19, p.10]. Correspondingly, the researcher conducts an in-depth analysis per case before comparing narratives between cases and extracting common themes [20]. IPA has been frequently applied in medical research [19] and when investigating women's experiences during menopause [e.g., 21, 22].

## Participants and sampling

We included n = 8 women from the placebo arm who experienced relief regarding their hot flush symptoms. Eligibility was based on the final RCT questionnaire; a score of 5 or higher on the overall improvement scale from 1 'very much worse' to 7 'very much better' was deemed as having experienced relief. The first 8 women who fulfilled this criterion were approached via phone, and they all agreed to participate. Thus, the experiences of women who did not respond to the placebo (n = 19; 38%) were not assessed. We opted for this selective focus and small sample size to (a) ensure a feasible analysis given the rich data generated and (b) because a homogeneous and small, purposively chosen sample is considered appropriate when applying the IPA method [20]. Our sample size aligns with the standard of IPA studies of including five to ten participants [19, 23, 24]. Before the interview, participants were informed about the study verbally and in writing and signed written informed consent.

## Data collection

Interviews were conducted in the treatment rooms of the Department of Psychosomatic Medicine, University Medical Centre Hamburg-Eppendorf, between March 2019 and February 2020. Each interview lasted 45–60 minutes, was audio-recorded and transcribed verbatim. Only the interviewer and the participant were present during the interview; personal data were omitted in the recording to ensure confidentiality. Field notes were taken after the interview. We created guiding questions a priori, assignable to one of two content blocks. Table 1 shows the seven guiding questions asked to all participants. The first block focused on menopausal symptom history, whereas the second block focused on women's experiences with the placebo

**Table 1. Guiding questions.**

| Block 1: Menopausal symptoms and treatments | |
| --- | --- |
| Q1 | When do your hot flushes bother you most? |
| Q2 | What comes to your mind when you have a hot flush? |
| Q3 | Could you please tell me a bit about your experiences with previous treatments? |
| **Block 2: Experiences with the OLP and the study** | |
| Q4 | What was your motivation for study participation? |
| Q5 | Why do you think the placebo worked for you? |
| Q6 | Could you tell me about how you perceived your hot flushes during the study? Were there any differences compared to before? |
| Q7 | What is your overall opinion of the study? |

treatment. Except for Block 1 preceding Block 2, there was no fixed question sequence as the interviews were unstructured. The interviewer was free to use prompts to explore interesting phenomena in-depth and, being aware of each woman's initial situation, enabled her to ask meaningful questions. The division into blocks served internal communication only. That is, for the participants, there was continuity throughout the interview. To keep participant effort to a minimum, transcripts and analysis results were not administered to participants for comment.

All interviews were conducted by MLF, a medical student who received qualitative research and IPA training before the interviews. When conducting and analyzing the interviews, she aspired to remain reflective about her pre-knowledge about menopause and placebo effects, her biomedical background, and her lack of personal experiences with menopause. Further information on the research team's characteristics is listed in S2 Appendix.

## Analysis

In line with the IPA protocol [20], interview transcripts were read multiple times; notes were taken. Afterward, themes and clusters (sub-ordinates themes) were extracted per subject and subsequently compared across subjects. When allocating citations to themes, they were adjusted or re-named if thereby the reading of the data became more differentiated and comprehensive. To increase the validity of the results, two authors (YP and MLF) read all interviews. With the research question in mind, we only extracted clusters and themes which contributed to understanding the participants' experiences with the OLP treatment. For instance, indications about "menopause linked to the end of fertility" or "stress due to unpredictability of flushing" were not included in the results. Correspondingly, the accounts of women in Block 2 formed the focus of our analyses. Nonetheless, the final themes and clusters reflect the data generated from all questions. MLF created the themes and clusters, critically examined by YP, followed by a thorough discussion between the authors YP, MLF, and TJK before establishing the final list of themes and clusters. Data saturation was achieved after the seventh interview; no new themes emerged with the eighth interview. Original quotations in German are provided in S3 Appendix.

## Results

An overview of the themes and clusters is provided in Table 2. Table 3 shows the participants' characteristics.

**Table 2. Clusters and themes.**

| Clusters | | Themes | |
|---|---|---|---|
| **A** | Openness, hope, and fascination | 1 | Openness: Hopeful but with low expectations |
| | | 2 | Curiosity |
| | | 3 | Placebos: Only gains, no losses |
| | | 4 | Lack of explanation & fascination for placebo effects go together |
| **B** | Motivational aspects | 5 | Seeking evidence for improvements |
| | | 6 | Attributions in favor of the treatment |
| **C** | Changes in symptom perception | 7 | Monitoring symptoms helped |
| | | 8 | Recognizing the influence of psychosocial factors |
| **D** | Empowerment | 9 | Control over symptoms |
| | | 10 | Agency |

**Table 3. Sample description.**

| Participant | OLP intake (weeks) | Symptom duration (years) | Previous hot flush treatments | Hot flush score (Frequency x intensity) [% change from baseline] | | |
|---|---|---|---|---|---|---|
| | | | | Baseline | Week 4 | Week 8 |
| **Andrea** | 4 | 4,5 | Black cohosh and other herbal remedies: no effects | 18.7 | 3.7 [-80.2%] | 1.9 [-89.8%] |
| **Meike** | 8 | 2 | Chaste tree: no effects, homeopathic salts: initially helpful | 15.9 | 6.7 [-57.9%] | 7.1 [-55.3%] |
| **Eva** | 4 | 2,5 | Hormone therapy: effective, homeopathic remedies: initially helpful | 9.3 | 4.7 [-49.5%] | 7.3 [-21.5%] |
| **Gabriele** | 8 | 4 | Chaste tree, homeopathy, black cohosh: no effects | 35.6 | 3.7 [-89.6%] | 6.9 [-80.6%] |
| **Ingrid** | 8 | 2,5 | Island moos: no effects | 20.7 | 10.4 [-49.8%] | 13.6 [-34.3%] |
| **Marlene** | 4 | 1,5 | none | 18.1 | 8.9 [-50.8%] | 6.7 [-63.0%] |
| **Edith** | 4 | 4 | Black cohosh, homeopathic salts: no effects | 28.6 | 18.9 [-33.9%] | 17.4 [-39.2%] |
| **Karin** | 4 | 4 | Sage tea: some effects | 18.2 | 3.4 [-81.3%] | 20.6 [+13.2%] |

*Notes*. OLP = Open-label placebo.

All names were changed. The hot flush score was the primary outcome of the preceding randomized-controlled trial. Each hot flush was to be rated as 1 = mild (hot flush, no sweating), 2 = moderate (hot flush and sweating), 3 = severe (hot flush and sweating, plus behavior to address the symptom). We only interviewed women who benefitted from the treatment. In the parent RCT, we found no differences between 4 or 8 weeks of intake.

## Cluster A–Openness, hope and fascination

**(1) Openness: Hopeful but with low expectations.** Women's attitudes at treatment start can be best described as open and come-what-may. They did not want to expect too much to protect themselves from possible disappointment. Nonetheless, they took part in the study and started taking the pills because they hoped the placebo would help alleviate their symptoms.

"I didn't expect anything in particular, but instead I thought, well, just give it a try. I approached it with some kind of openness and . . . yes, curiosity." (Marlene)

"Maybe because—I went in quite relaxed, so maybe I didn't have such high expectations. Oh, it has to work out now. I now have these . . . expectations. Instead I [. . .] thought: Gosh, either it works or it doesn't work. And if it doesn't work, then don't be disappointed, something like that." (Andrea)

"Well of course I was hoping for a success, but my expectations in the beginning I think weren't, well. . . not so. . .not so high." (Ingrid)

This attitude is consistent with the paradoxical nature of administering placebos, i.e., all women were honestly informed that they were taking part in a trial of placebos that have no active ingredients. Expecting benefits from pills without any active ingredients seemed implausible to these women. However, rational explanations may also serve to prevent disappointment, as laughter and rhetoric devices such as hyperboles ("all healed") show:

"I didn't expect anything. I was hoping. But I also thought, well, it's a placebo, right? What could possibly happen there after all (laughs)." (Gabriele)

"I approached it [. . .] without too much thinking. [. . .] Well, [I was expecting] also that it's gonna help me, sure, at least that they [the hot flushes] are a bit reduced, but—not such

high expectations. [. . .] It also wouldn't seem logical to me, that -ehm—I'd come out all healed." (Edith)

Women's hopes varied. Most had pragmatic attitudes and viewed improvements as nice-to-have.

"Well, in fact, I had hopes that these hot flushes would disappear, but my attitudes were technically neutral. [. . .] I didn't think, it will or won't help, but rather–Let's see what happens." (Karin)

Others had high hopes as they had tried several herbal remedies yet did not obtain any benefits.

"I thought, gosh, maybe you should give it a try, one more time, maybe for the last time, maybe it'll work. And if not then I would have–I guess–not done anything anymore [. . .]. Yes because then I somehow thought, gosh, well ok, then I'm probably one of those many women for which nothing works." (Andrea)

These yearnings for improvement were also reflected in the language (repetitions and exclamations underlined):

"Well let's just say consciously- unconsciously rather–ehm—I was crying for help. [. . .] then I thought, <u>ahhh</u>, <u>hopefully</u> I'll receive the placebos, yeah, there it was again, this -<u>hopefully, hopefully</u> [. . .] they will alleviate the hot flushes." (Edith)

All patients were aware of hormone therapy as a treatment, yet either decisively rejected the usage or considered it as a last resort option. For Meike, the willingness to initiate hormone therapy was viewed as an indicator of burden and desperation:

"I was actually going that far of thinking whether I should start with hormone therapy [. . .]. Because I just don't sleep anymore, period. I don't fricking sleep anymore and–ehm- this can't go on like this." (Meike)

She also viewed placebo research as cutting-edge, thus attaching high hopes to the treatment.

"Because you're moving into a completely new field and there's also a lot of hope attached to it, [. . .], so somehow something as big as that is also in the back of your head." (Meike)

For some participants, it is precisely their open and non-expectant attitudes that seemed to have contributed to symptom improvement. Because there was no obligation for the treatment to work, they were free to lay back and embrace whatever might happen, which was coupled with a general easing of pressure.

"This is why maybe they worked so well for me, I don't know. Because maybe I did not feel as tense and went in completely relaxed." (Andrea)

"If I was negative or something–approached it [the placebo intake] with a negative attitude–ehm- then I think it probably wouldn't have helped me. Or if I tried to control it." (Edith)

**(2) Curiosity.**   Women were inquisitive about what would happen given the contra-intuitive nature of the treatment.

"Yes, I found that extremely. . . interesting, that it really is just sugar." (Andrea)

"Things that you can't explain make you curious, right?" (Meike)

"I told all kinds of people about it–for sure, because I just found it super interesting." (Gabriele)

Except for Marlene, all women had experiences with CAM treatments (Table 3). However, they did not obtain satisfactory amelioration. Some women had clear expectations about CAM treatments, which contrasted with their attitudes towards the placebo treatment. Placebo improvements were attributed to its novelty effect.

"Hm well I think for the homeopathic salts there were expectations [. . .] because I've already made the experience that homeopathic salts can work [. . .]. And when I take homeopathic salts, I'm not curious." (Meike)

As positive expectations are commonly viewed as an efficacy booster, it was even more compelling that herbal remedies did not work, yet placebos did.

"The expectations [for herbal remedies] were definitely higher, and that's what makes it even more fascinating, that this has worked. That was really. . .well, cool." (Gabriele)

The curiosity went beyond the efficacy of the placebo treatment. For instance, Eva was wondering about dose-response relationships and speculated that taking the pills for eight weeks might have increased the effects.

"I would have thought like—the placebo would have even more effects, something like that, if you took it longer." (Eva)

**(3) Placebos: Only potential gains, zero losses.**   Participants considered placebos to be ideal as they cannot cause side effects. In contrast, other treatment options had drawbacks; hormone therapy was associated with harm, herbal treatments were not guaranteed to be efficacious yet required long-term out-of-pocket costs. Even though the placebo treatment would not necessarily result in improvements, it also came with no costs, at least not part of the study. Thus, women had "nothing to lose."

"Well that's really, I mean great, right? That you can obtain improvements without burdening the body with any substances. That's actually perfect." (Gabriele)

Women contrasted their open and positive attitudes towards the placebo already existent at study initiation to the general view of placebos being fake medicines for gullible individuals.

"Well, I think it's just because I have thought about it [the placebos] positively instead of thinking, gosh no, it's just a placebo. . . I think." (Eva)

"I don't believe in homeopathy and homeopathic salts, I think that's also rather a placebo effect (laughs). But I do believe in the placebo effect!" (Karin)

As menopause is not a disease, taking medications such as hormones was considered unfit. However, not taking any measures was also frustrating. By taking placebos, one could address the symptoms without harming the body.

"Hot flushes are not a disease, [. . .] and you don't have to immediately get the [chemical] clubs out, [. . .], but rather treating it a bit more naturally and not just ignoring it, yeah. This is something that I'd hope for, that this [message] comes through more." (Edith)

**(4) Lack of explanation and fascination for placebo effects go together.**   Some patients noted that it was a *"big mystery"* (Gabriele) why the placebo worked. OLP created a paradoxical conundrum. When inquired further, they reiterated the conditioning theory provided in the placebo briefing at the beginning of the RCT:

"Well [. . .] the body sees it a bit differently than the head [. . .] And that the body knows that pills are good for me, even if the head says there's nothing in there, something along those lines, I don't know. And that kind of made sense to me, so I won't think about it any further." (Gabriele)

As there was no easy rational explanation, benefits obtained through the placebo were viewed as absurd and comical. Interestingly, the indicated examples commonly involved a conversation with another person, hinting at the stigma and the fascination of responding to placebos, which was also discussed in prior OLP qualitative studies [13, 14].

"After I discontinued the pills, and when they [the hot flushes] have gotten more severe again, my husband said–Oh dear, you should have kept taking those placebos! (laughs)" (Karin)

"Well I also–ehm- had told my family beforehand that I was taking part, and I said: there are two groups, one group receives nothing, and the other group receives pills with nothing, like that. That's just what it was, and I think that's a good explanation. . .But that does show–I think quite clearly–how weird this all is (laughs), yes." (Gabriele).

The lack of a plausible explanation was considered a limitation. Conversely, the fact that placebos worked was associated with overall awe of what the body can do and that there are limitations to what humans can rationally explain.

"I know that there are tons of things [. . .] that we don't understand yet [. . .], but rather it's happening just on a physical level somehow [. . .]. The human being supposes that he can always understand what's going on. [. . .] But I believe in many things, and I could imagine that this played a role too, as I'm just very open for what's unlabeled." (Meike).

## Cluster B–Motivational aspects

**(5) Seeking evidence for improvement.**   Some women underlined their narratives of improvements with clear indicators, indicating that they closely monitored changes and possibly sought proof of benefit.

"Also, at night I don't have those big hot flushes anymore, for which I would have to wear a different shirt." (Andrea)

"I had […] more level one and less level three [hot flushes on the severity scale]." (Edith)

Validation of treatment benefits was also demonstrated in the language used during the interview (linguistic reinforcement underlined):

"Well, it <u>really was much less</u> […]. And previously, before the study [it] was also extreme and then later with the pills and also during the four weeks in which I didn't take any, well that was <u>clearly much better</u>." (Andrea)

Although the placebos aimed to reduce hot flushes, women also observed improvements in other menopausal symptoms.

"Even the mood, which also is–I mean during menopause you have certain moods, well, all that stuff around it were -ehm- less, definitely. […] Less heart racing, fewer panic attacks, […], or not at all, so it was really very diminished." (Edith)

"When I somehow couldn't effectively perform, or I would only come to work at 10am because I couldn't sleep all night, uhmm, because I had heart racing or something like that. I mean, I don't have that anymore, right? That's actually gone. I only have hot flushes–yet again—right now, yeah. And those depressed moods, I don't have them either at the moment." (Eva)

**(6) Attributions in favor of the treatment.**   As women wanted the treatment to work, they might have attributed improvements to the placebo yet sought alternative explanations for a lack of improvement.

"There were those couple days in which it increased a bit […], then I had the feeling that it's more my eating behavior and how I have slept and actually maybe external stressors on top […]–that these have caused one or two more hot flushes. And it's not because that the remedy didn't work, instead it's... those strains from other sources that affected me." (Marlene)

In some cases, multiple theories existed in parallel. For instance, Gabriele attributed her symptom reduction to both the placebo intake *and* the onset of her period, a time when she often experienced decreasing hot flushes. Similarly, Karin claimed that the placebos were helpful while also acknowledging the potential influence of the weather. Attribution could be tinged with doubt.

"Yes, the placebos also worked for me. Of course, I don't know whether it's the placebo and whether it is because that earlier it was summer and very hot […]. When I initiated the study, it was fall–and not so warm then." (Karin)

## Cluster C–Changes in symptom perception

**(7) Monitoring symptoms helped.**   Completing the hot flush diary was regarded as helpful in multiple ways. Some women felt validated in their symptom burden.

"Although I had it often, but still I didn't know how many these were- I mean I didn't count them. And to see that I actually did have 10 to 12 a day, that was also impressive to see, and

I thought–yeah, that was correct- the feeling that I have them all the time was correct." (Gabriele)

Seeing that their symptoms were not as severe as first thought, experiencing symptom relief through the placebo treatment, and observing this alleviation through the diary helped women feel less burdened.

"Well before [the study] I thought my hot flushes were severe–purely subjectively spoken [. . .]. Well, [during the study] I could somehow classify myself–ehm- better. During the course of the study, I simply didn't feel as . . .restricted [. . .] in my day-to-day [. . .] Definitely, this perception has clearly . . .clearly changed. Then I no longer found them as limiting or burdensome." (Ingrid)

Feeling improvement increased their hope and built anticipation for more improvement, potentially creating a positive cycle.

"It goes hand in hand. Well, I receive the validation that it's gotten a little better, and then I'm hopeful that it is the case, which gets validated again, and -ehm- maybe just attenuates my expectations of having a bad night a bit." (Meike)

"When I saw that these were just three or four–well that was really good (beaming) to see that it's actually really gotten less." (Gabriele)

The scientific context and the diary enabled some women to take on an observer perspective. Instead of being immersed in their burden, the distance created a sense of control.

"One has technically transitioned from being the sufferer to the bookkeeper" (Marlene)

"The fact alone [. . .] that you can explore it a bit scientifically through an exchange, you gain [. . .] a different perspective on it and maybe by that, a certain distance, [. . .] so you're not completely at its mercy." (Meike)

**(8) Recognizing the influence of psychosocial factors.**    Through study participation and the protocoling of symptoms, women were better able to spot psychosocial or behavioral triggers.

"Of course, I did -ehm- a bit of reading and then thought -mhm–what are some factors that could influence it [the hot flushes] positively, like diet, sports, enough sleep, relaxation, [. . .] and yes, for a while I also meditated. That also has a positive effect. Abstaining from spicy foods works good too. I recognized that better with this study [. . .]. The study has actually motivated me to observe it more closely. I haven't observed it that closely." (Marlene)

Stress was mentioned as an important aspect. By regulating their distress, women were able to exert influence on their symptoms. Decreased stress might have contributed to fewer hot flushes, given the established link between stress and hot flushes [25].

"Then during the day I just tried to take it slo- more relaxed, and to not stress myself out so much and that's when it's gotten a little bit better–so I felt–[. . .] well, I could regulate it a little." (Andrea)

"That became pretty clear [. . .] that, also during the intake, [. . .] I was under the impression that on the weekend for example, I had fewer hot flushes than during the week." (Ingrid)

### Cluster D–Empowerment

**(9) Control over symptoms.**  Knowing that they can fall back on placebo tablets, women reported a heightened sense of control. Edith compared the placebo tablets to *"clinging to cigarettes"*. Meike distinguished placebos from cognitive strategies such as mindfulness or distancing herself from worrying thoughts as a more tangible solution.

"Hormones etc. are also something that's concrete, and -ehm–it does make it easier for you when you have the feeling that, ok- here's something you can try, only that there are no side effects." (Meike)

Notably, no patient reported that their hot flushes had disappeared entirely. Instead, the experienced improvements under the placebo opened a window of opportunity for symptom acceptance. Knowing that symptoms are susceptible to change also provided mental relief and a sense of empowerment.

"I think it has just gotten easier to accept it, yeah. [. . .] I didn't have such severe–those super severe sleeping problems etc. [anymore], and because of that I was probably a bit clearer in my head I could also reflect a bit more about it [the hot flushes]." (Meike)

"The experience I guess, that- ehm- a certain influence is in some ways attainable and that made it easier for me." (Meike)

"And that's exactly that one thing, that for me is a VERY positive takeaway–I mean it strengthens your . . .your own powers and self-image, right? That a lot is in your own hands." (Marlene)

Andrea and Marlene experienced a slight recurrence of symptoms after discontinuation, yet they decided to take matters into their own hands.

"And then after I haven't taken the tablets for those four weeks, I was a bit under the impression that they [the hot flushes] came back and increased a bit. [. . .] But then I told myself, oh come on, just relax a bit. Don't get worked up about it all over again, something like that. And then it's gotten back down again. Well, that was pretty good." (Andrea)

Similarly, Marlene's attitude shines through her linguistic subtleties and positive affirmations: At treatment start, Marlene stated "that *it* will work", whereas, after treatment discontinuation, she noted, "*I* can do it. *I* was very optimistic."

This sense of self-efficacy enabled patients to tackle their symptoms beyond the study period. Andrea became more relaxed during the study and was able to maintain her benefits at the time of the interview (several months after treatment stopped). Gabriele purchased placebo pills from the pharmacy and planned on continuing intake in the upcoming months as they were still efficacious. Lastly, Marlene aimed to track her hot flushes to study further hot flush triggers.

**(10) Agency.**  Women regarded this study as an opportunity to do something against their hot flushes.

"Just by getting things rolling, coming here- literally, getting proactive, [. . .] I think that does something to you." (Meike)

Some women indicated that they always knew they "should" deal with their symptoms and the meaning of menopause in general. Women felt that the patient-clinician relationship and the interaction with the study team facilitated reflection and acceptance. Given that the patient-clinical engagement was similar in both RCT groups, this suggests that the clinical interaction is important -maybe even necessary- but not sufficient for OLP to work.

"I really thought it was really nice [. . .], the way she [the clinician] held the conversation, the questions she asked- ones where you had to really reflect upon, those that went beyond the surface, I liked that. And that of course also directed the perception towards a more . . . well objective level and helps with the acceptance." (Marlene)

"I HAVE to talk about things and here I could talk about it." (Eva)

Before the study, some participants tried to ignore their hot flushes as much as possible, fearing that *"giving it too much room"* (Marlene) would make matters worse. However, the study helped them develop a different *"inner attitude"* (Ingrid), i.e., by becoming proactive in accepting menopause as a natural part of life. Lastly, taking time to participate in the study was considered self-care.

". . .that I think I am being kind to myself, I am doing something for me." (Ingrid)

"I was really very very very happy [to participate] [. . .]. It's like, there is this one thing that no one can meddle with, it was my decision, and [. . .] I did it for myself. That was very important to me. A first step towards also learning this, yeah." (Edith).

## Discussion

From the interviews with menopausal women who benefitted from OLP treatment as part of a hot flush RCT, we obtained four subordinate themes describing women's experiences during the study: (A) Openness, hope, and fascination, (B) Motivational components, (C) Changes in symptom perception, and (D) Empowerment.

### Attitudes towards the placebo treatment: Hopeful, non-expectant, curious, fascinated, and at ease

All women in our interview already had a relatively positive view of placebos when entering the trial. The study was advertised as a placebo study, so only those open to placebos participated in the RCT. Participants hoped to benefit from the treatment and kept an open mind, yet did not dare to expect anything. Their understanding of expectations aligns with the definition of Rief & Petrie being "specific cognitions about the likelihood of future events". [26, p.602]. To them, the overall idea that placebos could work was plausible, yet the specific idea that the placebo would work *for them* was considered unlikely. Conceptually, hopes are associated with preference, whereas expectations are associated with probability [27]. OLP participants burdened by persistent symptoms appear to speak of hopes rather than expectations [28, 29]. The subjects in our study did not seem to be much different. These findings indicate that distinct positive expectations (operationalized as "I believe that I will get better after the treatment"), although a powerful predictor of placebo effects induced via expectation

manipulations in laboratory experiments [9, 30], may not be the decisive factor in the generation of OLP responses–a notion that has been backed up by the absence of any link between baseline expectations and subsequent OLP effects in our quantitative RCT [16] and those of other OLP trials [31–33]. Interestingly, according to predictive coding theories of placebo effects, for chronic conditions, uncertainty and imprecise expectations are considered advantageous prerequisites in creating a placebo response [9].

Although OLP treatments lack an active substance, most of our participants were motivated by exactly this characteristic: no active substance, no side effects. There was no pressure for the treatment to be effective ("Let's see what happens"), and knowing that harm cannot occur made the participants feel at ease. The seeming paradox of an inert pill creating measurable benefits made some women more curious about whether it would work and fascinated when it did. Nonetheless, the participants were aware that the idea of effective placebos was technically absurd, especially from an outside perspective, which corroborated the findings of two other qualitative OLP studies [29, 34].

Participants hypothesized that it would not have worked if they were dismissive of OLP. Their cautious hope aligned with findings from other qualitative studies examining patients' experiences with double-blind placebos or CAM treatments [35–37] and a recent qualitative study with IBS patients taking OLP or double-blind placebo [29]. Although a central role of despair has been suggested in previous OLP studies [28], most women in our study considered symptom alleviation as "nice-to-have". Our patients reserved the word "despair" or "desperation" for when they would seek hormone therapy. In the previous OLP studies, patients were refractory and generally tried multiple treatments without success. They were at the end of the rope. Presumably, because hormone therapy existed as a last-resort option, our patient population was less despairing. Two points can be inferred: Firstly, OLP trial participants are likely hopeful, yet hope can range from pragmatic to yearning. It appears that our participants were more pragmatic than other OLP patients. Secondly, unlike the common notion of placebos being better than nothing [34, 38], some women noted that placebos were perfect for hot flushes and, in that way, superior to hormone therapy. As menopause is not a disease, one should not use strong medications to "treat" the symptoms. However, not taking medications was also viewed as an unsatisfying option. Placebos constitute an optimal middle path, as one proactively addresses the symptoms without harming the body.

The hopes of experiencing symptom relief and the curiosity surrounding placebos may have facilitated subsequent psychological processes such as increased attention towards bodily signals and attribution in favor of the treatment. For instance, women noted changes in "objective" indicators of the OLP response, like less sweaty shirts. Some women attributed slight aggravations during the treatment to external factors such as a bad night's sleep or warm weather, further pointing to the role of motivational factors. "Curiosity" may also be related to Bayesian brain theories of placebo, which emphasize "surprisal" and searching for the "difference that makes a difference" [9]. In a series of experimental studies with healthy participants, Geers and colleagues found that symptom reduction under placebo was largest when this reduction was in line with participants' motivation and when study subjects focused on their bodily processes [39, 40]. Although it is unknown whether these laboratory studies can be translated into clinical practice, the hopes and related motivation to get better seem to play a role in women's experiences during intake.

## Changes in symptom perception helped patients to regain control

Biopsychosocial models of menopause suggest that environmental, cognitive, emotional, and behavioral factors can modulate symptoms [41]. From the interviews, we learned that many

women came to perceive their symptoms slightly differently. In specific, using the diary to monitor symptoms helped. By seeing the actual numbers, some felt validated in how they had perceived their symptoms. Others were able to spot lifestyle-related predecessors better. Moreover, the diary provided feedback on improvements. It is worth noting that this self-evaluation process was unlikely to be the significant cause of OLP improvement as the no-treatment control group underwent the same procedures.

Closely observing the symptoms helped women ascertain the role of stress and gain a neutral distance. Moreover, knowing that their hot flushes *can* ameliorate provided patients with a sense of control and self-efficacy. Participating in the study and taking placebos meant becoming proactive in addressing one's symptoms and taking care of oneself, which empowered these women. IBS patients who underwent the OLP treatment also reported that being proactive might have played a positive role [29]. A recent psychological analysis of OLP patients describes how the paradoxical conundrum of OLP has an implicit message of self-healing, empowerment and agency [42]. The essential role of perceived control was pointed out in numerous studies, e.g., investigating CAM treatment responses across several conditions [43, 44], including women in menopause [45, 46]. Further, women who took part in a group CBT against hot flushes marked "regaining control" as the main driver of their obtained benefits [21, 47].

To our knowledge, there have been two qualitative studies, each of both OLP and OLP plus conditioning (C+OLP). An OLP study with chronic IBS patients is most similar to ours in methodology and findings. Like our study, IBS patients spoke of hope, were reluctant to attribute improvement to placebo, engaged in self-reflection, and felt empowered [29]. Another qualitative study followed an experimental OLP pain trial comparing two different OLP rationales [34]. This high-quality study assessed how participants conceptualize the placebo effect, their thoughts on OLP, and underlying mechanisms concerning the rationale and their experiences. Two C+OLP studies sought to reduce medication dosages required for the treatment. One study paired amphetamine salts with placebo pills in pediatric attention deficit hyperactivity disorder and found that patients and their parents had low expectations and were supportive of the approach but mostly neutral or doubtful of its effects. Yet, they nonetheless improved with less medication and C+OLP [48]. The second C+OLP study was a pilot observational study that found patients suffering from acute pain and taking opioids were comfortable with the C+OLP rationale, thereby supporting the feasibility of a future larger trial [49].

## Limitations

As we purposively included a homogenous sample, i.e., only women who had improved under the placebo treatment, we cannot compare our results to women who had not responded to the placebo treatment. For instance, Haas et al. found that OLP participants had flexibility in thinking, i.e., they attributed an ineffective treatment to the pills being inert [29]. However, when the treatment did work, they had other explanations, such as becoming proactive about their situation. Naturally, we could not distill findings such as these with our limited focus. Nonetheless, our study's distinction lies in its rich per-case data and the possibility of interpreting patients' experiences in context.

Furthermore, we did not interview patients in the no-treatment control who had similar patient-clinician relationships with the team, identical self-monitoring, and likely analogous life stresses and reflections on menopause. We suspect that, had we interviewed patients in the no-treatment control, it would have re-enforced the clear conclusion of our quantitative study-OLP made the difference. Given the conclusion of our quantitative study, we speculate that some of the patient-reported experiences on OLP, such as the support of staff and changes

in symptom perception, were necessary components of OLP but not sufficient in themselves for symptom improvement.

Another limitation is the participants' awareness of the investigators' hypothesis. During the informed consent session of the RCT, all trial participants were informed about the study's aim of investigating OLP efficacy for hot flushes. Although only one of eight interviewed women knew the interviewer from the RCT, we cannot rule out that the benefits of OLP or the study were accentuated towards the interviewer to please the researchers. However, we considered these factors when interpreting the data. Furthermore, it is hard to give credence to the concern of pleasing the researchers given the compelling positive quantitative results that our RCT recorded. If it were a pharmaceutical or procedural RCT, such concerns would not be discussed.

## Conclusion

This is the first qualitative study of OLP for menopausal hot flushes. The patient interviews with menopausal women who benefitted from OLP treatment as part of an RCT indicate that placebos administered without deception and embedded in a clinical context can elicit salubrious benefits. Participants noted that they had an initially positive, curious, and "couldn't hurt" attitude towards placebos. They hoped their symptoms would get better but refrained from expecting improvements. By monitoring the symptoms closely, the participants perceived their symptoms differently, felt reassured in their burden, and, by taking agency, in more control over their symptoms. Considering these experiences can help us focus on the factors most relevant to patients and refine treatments to increase patient satisfaction and optimize outcomes.

## Supporting information

**S1 Appendix. Placebo rationale.**
(DOCX)

**S2 Appendix. Research team description.**
(DOCX)

**S3 Appendix. Original quotations in German (with context information and English translation).**
(DOCX)

## Acknowledgments

The authors thank all participating women for providing us with rich information about their intake experience.

## Author Contributions

**Conceptualization:** Yiqi Pan, Ted J. Kaptchuk, Yvonne Nestoriuc.

**Formal analysis:** Yiqi Pan, Miriam L. Frank.

**Funding acquisition:** Yvonne Nestoriuc.

**Investigation:** Miriam L. Frank.

**Methodology:** Yiqi Pan, Miriam L. Frank.

**Project administration:** Yiqi Pan, Miriam L. Frank.

**Supervision:** Ted J. Kaptchuk, Yvonne Nestoriuc.

**Validation:** Miriam L. Frank.

**Visualization:** Yiqi Pan.

**Writing – original draft:** Yiqi Pan.

**Writing – review & editing:** Miriam L. Frank, Ted J. Kaptchuk, Yvonne Nestoriuc.

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
