## [Decision Letter · Decision Letter 0]

28 Apr 2022

PONE-D-21-27340"Let's See What Happens:" - Women's experiences of open-label placebo treatment for menopausal hot flushes in a randomized-controlled trialPLOS ONE

Dear Dr. Pan,

Thank you for submitting your manuscript to PLOS ONE. After careful consideration, we feel that it has merit but does not fully meet PLOS ONE’s publication criteria as it currently stands. Therefore, we invite you to submit a revised version of the manuscript that addresses the points raised during the review process.

The manuscript has been evaluated by two reviewers, and their comments are available below.

The reviewers have raised a number of major concerns. They feel the manuscript they request improvements to the reporting of methodological aspects of the study. The reviewers also note concerns about the statistical analyses presented.

Could you please carefully revise the manuscript to address all comments raised?

We look forward to receiving your revised manuscript.

Kind regards,

Thomas Phillips, PhD

Staff Editor

PLOS ONE

Journal Requirements:

2. Please ensure you have included the registration number for the clinical trial referenced in the manuscript.

3. When reporting the results of qualitative research, we suggest consulting the COREQ guidelines  or other relevant checklists listed by the Equator Network, such as the SRQR, to ensure complete reporting (http://journals.plos.org/plosone/s/submission-guidelines#loc-qualitative-research). Moreover, please provide the interview guide used as a Supplementary File.

Reviewers' comments:

Reviewer's Responses to Questions

**Comments to the Author**

1. Is the manuscript technically sound, and do the data support the conclusions?

Reviewer #1: Yes

Reviewer #2: Partly

2. Has the statistical analysis been performed appropriately and rigorously? 

Reviewer #1: N/A

Reviewer #2: Yes

3. Have the authors made all data underlying the findings in their manuscript fully available?

Reviewer #1: No

Reviewer #2: No

4. Is the manuscript presented in an intelligible fashion and written in standard English?

Reviewer #1: Yes

Reviewer #2: Yes

5. Review Comments to the Author

Reviewer #1: The manuscript is very well written, appears thoroughly prepared and contains crucial contribution to the field of OLP and placebo research. The methods are precisely described and the results are very interesting.

I have some remarks:

- There are references lacking in the introduction (e.g., 34, 37, ff).

- Line 50: I would not write „recently“. It is not so recent anymore.

- You might think about omitting the p values of your study in the intro for better readability

- It would be helpful to have the verbal script of the OLP treatment rationale in the appendix or a link

- Please provide also the a priori defined questions for the two blocks. It would be interesting to know, how many of these questions were asked in the interviews.

- Further, original comments in German might be added to the supplement.

- You could omit to mention the RCT results again in the methods section.

- Regarding the potential bias for only interviewing women who showed improvement: You may make this transparent already in the abstract: “women who benefited…”

- Table 1 might be adapted: Substitute the changed names with the N of participants who mentioned the theme. This would give a better overview and enable more focus on the themes. The themes could also be highlighted (e.g., in bold) and the letters A-D in a first, separate column.

- Sociodemographic data of the qualitative sample is missing. Please add descriptive information.

- Did you correct the verbatim grammatically? The statements seem very structured.

Small correction suggestions:

. line 52 e.g., 13 RCTs

. , at least not part of the study

. line 422 placebos being “better than nothing” treatments

Reviewer #2: Abstract

line 21: It is not true that no study investigated patients’ experiences undergoing treatment. There are now at least two studies who did so:

Bernstein MH, Magill M, Weiss A-P, et al. Are conditioned open placebos feasible as an adjunctive treatment to opioids? results from a single-group dose-extender pilot study with acute pain patients. Psychother Psychosom 2019;88:380–2.

Locher, C., Buergler, S., Nascimento, A. F., Kost, L., Blease, C., & Gaab, J. (2021). Lay perspectives of the open-label placebo rationale: a qualitative study of participants in an experimental trial. BMJ open, 11(8), e053346.

line 28: the summary of the abstract seems very optimistic, considering that many patients had low expectations about the placebo treatment. I would urge that the authors list pro and cons of the approach

Introduction

line 38: the authors state that recent research into the neurobiological underpinnings of placebo effects have been conducted. I would prefer a list of references after this sentence.

line 58: again, it is not true that no study has yet assessed participants’ experiences taking OLP (Bernstein et al.; Locher et al; see abstract comment). I would recommend that the authors briefly discuss the findings of these qualitative studies and outline how their approached differed from them.

Methods

lines 76: the original RCT included 100 women suffering from hot flushes. The nested qualitative study only conducted interviews with a total of 8 patients. This is a very small sample size, also for a qualitative approach.

line 99: I am also very worried that the authors only included patients with an experience of relief in the RCT (i.e., a score of 5 or higher on the overall improvement scale [range from 1-7] was mandatory in order to take part in the qualitative study). I think it would have been much more representative if participants with and without symptom relief would have been eligible. This would allow researchers to learn from both subgroups and to have an in-depth insight about chances and hurdles of the OLP approach.

line 11: the authors designed two blocks for the unstructured interviews, whereby the first block focused on symptom history. The authors state that the second block, which concentrated on placebos, was the focus of the current analysis. Why did the authors create two blocks and then only focused on one of them? I would advise that both blocks are considered equally important for the results, especially with the claim that these two blocks interact (line 116).

Results

Table 1: all themes listed in table 1 are positive attitudes towards the OLP approach. This is surprising since the authors also state that “almost all women reported that they had little expectations”. I would urge that the authors build an independent category for this statement and also list it in table 1. A statement like “I approached it without too much thinking. It also wouldn’t seem logical to me, that I’d come out all healed” was listed under the main category “openness and hope”. This is not at all intuitive to me.

line 252: in one of the citations, linguistic reinforcement was underlined by the authors. I would recommend to do this consistently throughout all citations (especially because the authors did not apply a content/thematic analysis that only concentrates on the content of the quotations but rather more a phenomenological approach that is interested in the lived experiences of participants)

Discussion

line 392: the authors state that participants were hopeful, curious, fascinated, and at ease. Again, I think that low expectations should also be discussed. The finding that all women had a relatively positive view of placebos when entering the trial could be associated with the fact that they all experienced symptom relief (i.e., which could have been the cause of a positive attitude).

6. PLOS authors have the option to publish the peer review history of their article (what does this mean?). If published, this will include your full peer review and any attached files.

Reviewer #1: **Yes: **Antje Frey Nascimento

Reviewer #2: **Yes: **Cosima Locher

---

## [Author Response · Author response to Decision Letter 0]

27 Jun 2022

Dear reviewers, please find our response to your excellent raised points in the attached "Response to Reviewer" file.

---

## [Decision Letter · Decision Letter 1]

12 Sep 2022

PONE-D-21-27340R1"Let's See What Happens:" - Women's experiences of open-label placebo treatment for menopausal hot flushes in a randomized-controlled trialPLOS ONE

Dear Dr. Pan,

Thank you for submitting your manuscript to PLOS ONE. After careful consideration, we feel that it has merit but does not fully meet PLOS ONE’s publication criteria as it currently stands. Therefore, we invite you to submit a revised version of the manuscript that addresses the points raised during the review process.

We look forward to receiving your revised manuscript.

Kind regards,

Walid Kamal Abdelbasset, Ph.D.

Academic Editor

PLOS ONE

Journal Requirements:

Reviewers' comments:

Reviewer's Responses to Questions

**Comments to the Author**

1. If the authors have adequately addressed your comments raised in a previous round of review and you feel that this manuscript is now acceptable for publication, you may indicate that here to bypass the “Comments to the Author” section, enter your conflict of interest statement in the “Confidential to Editor” section, and submit your "Accept" recommendation.

Reviewer #1: All comments have been addressed

Reviewer #3: (No Response)

2. Is the manuscript technically sound, and do the data support the conclusions?

Reviewer #1: Yes

Reviewer #3: Partly

3. Has the statistical analysis been performed appropriately and rigorously? 

Reviewer #1: N/A

Reviewer #3: I Don't Know

4. Have the authors made all data underlying the findings in their manuscript fully available?

Reviewer #1: Yes

Reviewer #3: Yes

5. Is the manuscript presented in an intelligible fashion and written in standard English?

Reviewer #1: Yes

Reviewer #3: Yes

6. Review Comments to the Author

Reviewer #1: I thank the authors for considering my suggestions and addressing my concerns.

The authors may consider excluding a remaining opening bracket in line 94 and change: „Was“ instead of „What“ in Table 1 in Q2.

Reviewer #3: TITLE: This particular study where 8 women where interviewed after administering a placebo is supposed to be a randomized control trial however, it is not clear when the randomization was done hence the title seems to contradict the study design.

I9NTRODUCTION: The 1st sentence in page 4 is not clear, did the placebos significantly reduce or increase menopause related quality of life?

METHODS: The 1st sentence which states states that the study is the second part of a design is not clear and this is not referenced? In line 77, the quantitative study referred to is not referenced and no information or synopsis is given for better understanding . In line 78, what do the authors mean by -- moderate subjective burden? In line 79, what do the authors mean by --- one week run in to access base line symptoms and what are these symptoms? In line 88, COREQ should 1st be written in full. Line 89, which protocol are the authors referring to and how is it related to this study?

7. PLOS authors have the option to publish the peer review history of their article (what does this mean?). If published, this will include your full peer review and any attached files.

Reviewer #1: **Yes: **Antje Frey Nascimento

Reviewer #3: No

---

## [Author Response · Author response to Decision Letter 1]

21 Sep 2022

Dear reviewers, 

Thank you for taking the time to review our manuscript! 

We were happy to incorporate your feedback in the revised version. Please note that all line references refer to the marked-up version. 

With my kind regards and on behalf of all authors, 

Yiqi Pan 

-----

Reviewer #1: 

I thank the authors for considering my suggestions and addressing my concerns.

The authors may consider excluding a remaining opening bracket in line 94 and change: „Was“ instead of „What“ in Table 1 in Q2.

A: Wonderful, thank you! We have changed the typo in Table 1 and removed the open bracket in (what is now) line 72. 

------

Reviewer #3: 

TITLE: This particular study where 8 women where interviewed after administering a placebo is supposed to be a randomized control trial however, it is not clear when the randomization was done hence the title seems to contradict the study design.

A: Thank you for raising this issue. We have added a specification in line 60 (italic): “In this study, we investigated the experiences of menopausal women who underwent and benefitted from OLP treatment for hot flushes that was administered as part of an RCT.” We hope this specification early in the manuscript will guide the reader’s understanding of the study’s design. 

Moreover, we have added that the eight women we’ve approached and included in this qualitative study are the first eight women who have improved in the quantitative study (page 5, line 107f.)

I9NTRODUCTION: The 1st sentence in page 4 is not clear, did the placebos significantly reduce or increase menopause related quality of life?

A: Thank you for pointing this out. We corrected the sentence (page 4, line 70).

METHODS: The 1st sentence which states states that the study is the second part of a design is not clear and this is not referenced? 

In line 77, the quantitative study referred to is not referenced and no information or synopsis is given for better understanding . 

A: Excellent point – we have now added the references for the study protocol and the results of the RCT. To clarify the “explanatory sequential mixed-methods design” (line 75) that we’ve used, we added a sentence on page 4, line 75ff (“Details on the design of the preceding quantitative study, i.e., the RCT, are provided in the study protocol [17] and the RCT results publication [16].”). 

A very brief summary of the RCT’s results is provided in the introduction (lines 69f.), and a synopsis of the RCT’s design is given in the “Design” paragraph (lines 77 – 89).

In line 78, what do the authors mean by -- moderate subjective burden? In line 79, what do the authors mean by --- one week run in to access base line symptoms and what are these symptoms? 

A: We agree that these terms are unclear and have done some rewrites (lines 77 – 81): “In brief, 100 women in peri- or post-menopause who had at least five hot flushes per day and were moderately or severely burdened by their symptoms were randomized to receive four weeks of OLP or no treatment. Before the randomization, all participants protocoled their hot flushes for a week (baseline assessment).”

In line 88, COREQ should 1st be written in full. 

A: We’ve added the name in full (line 91). 

Line 89, which protocol are the authors referring to and how is it related to this study?

A: We refer to the study protocol and have added the word “study” (line 93) for further clarity.

---

## [Decision Letter · Decision Letter 2]

10 Oct 2022

"Let's See What Happens:" - Women's experiences of open-label placebo treatment for menopausal hot flushes in a randomized-controlled trial

PONE-D-21-27340R2

Dear Dr. Pan,

We’re pleased to inform you that your manuscript has been judged scientifically suitable for publication and will be formally accepted for publication once it meets all outstanding technical requirements.

Kind regards,

Walid Kamal Abdelbasset, Ph.D.

Academic Editor

PLOS ONE

Additional Editor Comments (optional):

Reviewers' comments:

Reviewer's Responses to Questions

**Comments to the Author**

1. If the authors have adequately addressed your comments raised in a previous round of review and you feel that this manuscript is now acceptable for publication, you may indicate that here to bypass the “Comments to the Author” section, enter your conflict of interest statement in the “Confidential to Editor” section, and submit your "Accept" recommendation.

Reviewer #1: All comments have been addressed

Reviewer #3: All comments have been addressed

2. Is the manuscript technically sound, and do the data support the conclusions?

Reviewer #1: Yes

Reviewer #3: Yes

3. Has the statistical analysis been performed appropriately and rigorously? 

Reviewer #1: N/A

Reviewer #3: I Don't Know

4. Have the authors made all data underlying the findings in their manuscript fully available?

Reviewer #1: No

Reviewer #3: Yes

5. Is the manuscript presented in an intelligible fashion and written in standard English?

Reviewer #1: Yes

Reviewer #3: Yes

6. Review Comments to the Author

Reviewer #1: I have nothing to add. This is a very interesting study and it is valuable to see the subjective explanations of the women who underwent the OLP intervention in addition to the results of the RCT!

Reviewer #3: (No Response)

7. PLOS authors have the option to publish the peer review history of their article (what does this mean?). If published, this will include your full peer review and any attached files.

Reviewer #1: **Yes: **Antje Frey Nascimento

Reviewer #3: No
